# Treatment of Esophago–Airway Fistula after Esophageal Resection

**DOI:** 10.3390/healthcare11243165

**Published:** 2023-12-14

**Authors:** Janusz Włodarczyk, Tomasz Smęder, Piotr Obarski, Mirosław Ziętkiewicz

**Affiliations:** 1Department of Thoracic and Surgical Oncology, Jagiellonian University Collegium Medicum, John Paul II Hospital, 31-202 Cracow, Poland; 2Department of Thoracic and Surgical Oncology, John Paul II Hospital, 31-202 Cracow, Poland; tomasz.smeder@gmail.com (T.S.); piotr.obarski@gmail.com (P.O.); 3Department of Anesthesiology and Intensive Care, Jagiellonian University Collegium Medicum, John Paul II Hospital, 31-202 Cracow, Poland; mj.zietkiewicz@gmail.com

**Keywords:** esophageal carcinoma, esophageal resection, esophago–airway fistula

## Abstract

(1) Background: Esophago–airway fistula after esophageal resection is a rare, life-threatening complication associated with a high postoperative mortality rate. Managing this condition is challenging, and the prognosis for patients is uncertain. The results and our own approach to treatment are presented. (2) Material and Methods: We present a retrospective analysis of a group of 22 patients treated for an esophago–airway fistula between 2012 and 2022, with 21 cases after esophageal resection and one during the course of Hodgkin’s disease. (3) Results: Twenty-two patients were treated for an esophago–airway fistula. Among them, a tracheobronchial fistula occurred in 21 (95.4%) patients during the postoperative period, while 1 (4.5%) was treated for Hodgkin’s disease. Of these cases, 17 (70.7%) patients underwent esophageal diversion with various treatments, including intercostal flap in most cases, greater omentum in one (4.5%), latissimus dorsi muscle in two (9%), and greater pectoral muscle in one (4.5%). Esophageal stenting was performed in two patients (9.0%), and one (4.5%) was treated conservatively. Unfortunately, one patient (4.5%) died after being treated with bronchial stenting, and two (9.5%) experienced a recurrence of the fistula. (4) Conclusions: The occurrence of an esophago–airway fistula after esophagectomy is a rare but life-threatening complication with an uncertain prognosis that results in several serious perioperative sequelae.

## 1. Introduction

The occurrence of an esophago–airway fistula (EAF) after esophagectomy is influenced by the anatomical relationship between the esophagus and the bronchial tree. In patients who have undergone esophageal resection, these fistulas are typically found between the gastric conduit and the bronchial tree. Additionally, they may occur between the bronchial tree and the lung parenchyma. The procedure is challenging, and the approach to management depends on factors such as the fistula location, graft vascularization, and the patient’s clinical condition. Treatment methods may involve primary fistula repair, reconstruction using pedicled or free extra or intrathoracic muscle flap, esophageal diversion followed by reconstruction, or stenting. Unfortunately, despite advanced management, the prognosis for patients remains uncertain [1,2,3]. An EAF, a rare complication following esophageal resection occurring in approximately 3% of cases, presents an immediate life-threatening risk to the patient [3].

The current literature typically discusses small groups of treated patients, proposes various techniques, and often, the treatment decision depends on the center’s experience [1,2,3].

The aim of our paper is to share our own experiences, present our results, and propose a possible approach to treatment. We provide insights based on a large group of patients successfully treated, resulting in a low perioperative mortality rate.

## 2. Material and Methods

A retrospective study was conducted to evaluate 630 patients who underwent esophagectomy between the years 2012 and 2022 at Jagiellonian University’s Department of Thoracic Surgery in Cracow, Poland. We performed a retrospective analysis of the patients’ charts (Figure 1).

The stage of esophageal carcinoma was determined based on the UICC classification [4]. Patients with squamous cell carcinoma routinely underwent preoperative chemo-radiotherapy according to the CROSS protocol: 41.4 Gy plus carboplatin/paclitaxel, while those with adenocarcinoma (EGJ) received chemotherapy according to the FLOT protocol: 5-FU/leucovorin/oxaliplatin/docetaxel. 

### 2.1. Inclusion and Exclusion Criteria

#### 2.1.1. Patient-Inclusion Criteria

Patients qualified for radical treatment of esophageal cancer according to the CROSS or FLOT protocol and confirmed endoscopically (EUS, EBUS) with no invasion into the bronchial tree [5,6].Patients after radical esophagectomy R0.Patients with an endoscopically confirmed EAF.Patients undergoing chemoradiotherapy for a cancer other than esophageal cancer and esophageal surgery resulting in an EAF.

#### 2.1.2. Patient-Exclusion Criteria

Patients after non-radical esophageal resection with an R1 feature.Patients after intraoperative injury to the bronchial tree.Patients with confirmed leakage only in the esophagogastric anastomosis.Patients with a preoperatively confirmed EAF in the course of esophageal cancer.Severe patient condition disqualifying them from surgical treatment.

In the postoperative period, all patients underwent endoscopic examination to determine the integrity of the esophago-gastric anastomosis. When there was suspicion of an EAF, bronchoscopy was conducted, and bronchial secretions were collected for microbiological examination. During the surgery, fluid samples were collected from the pleural cavity for microbiological analysis. Subsequently, all patients received antibiotic therapy based on the results of the microbiological examination. Before the planned surgical treatment, patients underwent chest tomography (CT scan), esophagoscopy, and bronchoscopy.

We classified the occurrence of EAF based on our own classification, which is presented below. Patients diagnosed with Type II fistula without signs of ischemia in the esophageal graft were considered candidates for re-anastomosis and fistula repair in the bronchial tree using intercostal muscle flap, pectoral or latissimus dorsi muscles, or the insertion of biomaterial. For patients with type III fistula and signs of esophageal graft ischemia, myoplasty of the bronchial tree or biomaterial insertion, along with esophageal diversion to the neck, was the preferred approach.

Sepsis was defined according to the Surviving Sepsis Campaign as life-threatening organ dysfunction caused by a dysregulated host response to infection [7].

Mediastinitis has been defined as the presence of infected content in the mediastinum in the form of fluid and/or abscess, and/or necrotic tissues in the posterior mediastinum. A classification has been adopted distinguishing limited mediastinitis associated with type II fistula and progressive (diffuse) mediastinitis associated with type III fistula.

The restoration of gastrointestinal continuity was performed after 3 months, achieving the stabilization of the patients’ general condition. Until the time of surgical treatment, patients remained on enteral nutrition. For the purpose of this report, a patient’s quality of life after completing the surgical treatment was broadly defined by assessing the patient’s ability to nourish him- or herself. A good outcome was defined as the ability eat while meeting the energy requirements, while a poor outcome was defined as the inability for full oral nutrition or the inability to restore gastrointestinal continuity.

The jejunostomy was routinely removed approximately 1 month after restoring gastrointestinal continuity. In case of difficulties with oral nutrition or recurrence of the fistula, it was left in place, and enteral nutrition was maintained.

The following data were analyzed: demographic information, the American Society of Anesthesiologists (ASA) classification, and length of stay in the intensive care unit.

The occurrence of an EAF was assessed based on the following parameters: the impact of preoperative radiochemotherapy, the length of esophageal tumor infiltration, the number of removed lymph nodes, and the duration of surgery.

### 2.2. Surgical Technique

#### 2.2.1. Open Esophageal Resection (OER)

After opening the abdominal cavity, a gastric tube with a gastric omentum flap (GOF) was prepared followed by celiac trunk lymphadenectomy and placement of a nutritional microjejunostomy. Through a right thoracotomy, the esophagus was visualized by gaining access over the 6th rib without cutting the latissimus dorsi muscle (LDM). The azygos vein was routinely resected. Dissection of the esophagus was performed and the mediastinal lymph nodes were removed, including those located around the esophagus, tracheal bifurcation, right and left bronchi, and upper mediastinum between the trachea and superior vena cava in the region of the right laryngeal nerve. This allowed visualization of the left laryngeal nerve above the aortic arch. A gastric tube was then passed either retrosternally or through the posterior mediastinum. The anastomosis was performed within the thoracic apex or neck in patients with squamous cell carcinoma (SCC) and covered with the GOF. Cervical lymphadenectomy was performed routinely. An ultrasonic knife, Enseal bipolar device, and titanium clips were used during the surgery. On the 7th postoperative day, an endoscopic assessment of anastomosis tightness was performed.

#### 2.2.2. Esophageal Resection Using a Minimally Invasive Technique (MIT)

In patients undergoing the MIT, 5 ports were inserted into the peritoneal cavity. The gastric tube was prepared using an endostapler, and a celiac trunk lymphadenectomy was performed. The patient was then placed in the left decubitus position. Four ports were made and an esophageal resection with mediastinal lymphadenectomy was performed. An anastomosis was created using a circular stapler (Orvil, Covidien, Mineapolis, MN, USA). An ultrasonic knife was used during surgery.

#### 2.2.3. Intraoperative Ventilation

During esophageal resection, patients were intubated with a double-lumen left-sided tube (Sumi, Poland) to allow one lung ventilation and isolation of the non-dependent lung on the operative side. Pressures in the tracheal and bronchial balloons were routinely checked with a manometer. Ventilation was performed using the pressure-controlled mode (PCV) with small tidal volumes of ≤6 mL/kg ideal body weight (IBW). PEEP was set at a default level of 5 cm H_2_O and adjusted accordingly. FiO_2_ was kept at the lowest effective level to prevent the damaging and toxic effects of oxygen and keep SpO_2_/SaO_2_ above 94%. Manual repetitive alveolar recruitment maneuvers were used to prevent atelectasis. 

### 2.3. Surgical Treatment

#### Patients with Postoperative EAF

Primary repair was performed in patients with an anastomotic leak of no more than 2 cm and no macroscopic evidence of gastric conduit necrosis. A fistula in the tracheo-bronchial tree (TBT) was repaired when it was no larger than 2 cm. The suture line was covered by an intercostal muscle flap (IMF) or latissimus dorsi muscle (LDM), GOF, and a conduit.Patients who did not meet the criteria for primary repair were eligible for esophageal diversion and bronchial tree fistuloplasty. A fistuloplasty was performed using either an IMF, pectoralis major muscle (PMM), LDM, GOF, or a biomaterial.Conservative treatment was initiated in patients who had a fistula of up to 1 cm in size, no pleural fluid on CT scan, distended pulmonary parenchyma without pneumothorax, air leakage, and a gastric graft covering the fistula. All patients were started on broad-spectrum antibiotic therapy. Control bronchoscopies were performed on postoperative days 1, 3, 5, and 7.Use of TBT or esophageal stents were used in patients whose clinical condition prohibited surgical treatment or supportive treatment was ineffective.

### 2.4. IMF Technique

The IMF was dissected about 15 cm above the 5th rib at the level of the tracheal bifurcation. The bronchi were repaired using PDS 000 single sutures (Ethicon, Cincinnati, OH, USA). PDS 000 sutures were placed on the free edge of the intercostal muscle to avoid damage to its vascularization. The bronchial wall was then punctured close to the fistula from the outside to the inside, then approximately 2 mm from the puncture, a suture was pierced outward, and the muscle was punctured from the inside to the outside. After this suture was placed, the muscle was slipped over the injured bronchus and the sutures were tied. Three to four stitches were placed (Figure 2). The 4th rib was resected. The tightness of the repair was confirmed using the water test.

### 2.5. Fistula Greater Than 2 cm—Technique

In patients with a tracheobronchial wall defect greater than 2 cm, a biomaterial, LDM, or GOF was used. The biomaterial Permacol (Covidien, Mineapolis, MN, USA) was used and was sewn into the fistula using single or running sutures and then reinforced with an IMF using interrupted PDS 000 sutures.

Surgical complications were assessed according to the Clavien–Dindo scale [8]. 

The following division of fistulas was adopted:Fistula after intraoperative repair of a tracheobronchial tree (TBT) injuryEsophageal anastomotic leak with a fistula in the TBT without conduit necrosisEsophageal anastomotic leak with a fistula in the TBT with conduit necrosisFistula in the TBT without evidence of an esophageal anastomotic leakLung parenchyma fistula with or without evidence of an esophageal anastomotic leak

#### Follow-Up

Patients were subject to post-operative check-ups every 3 months in the first year and every 6 months in subsequent years. During the check-ups, routine assessments included chest imaging such as chest tomography, as well as endoscopic examinations (gastro-scopy), endoscopic ultrasound with fine needle aspiration (EUS-FNA), endobronchial ultrasound with transbronchial needle aspiration (EBUS-TBNA) with microscopic verification, and PET-CT scans. During the follow-up visit, the patients’ general condition, dysphagia, and dyspnea were assessed. If a follow-up check on site was not feasible, patients were interviewed by phone.

### 2.6. Statistical Analysis

Cox proportional hazards regression models was used to assess the impact of chemo-radiotherapy as a potential risk factor for the occurrence of EAF after esophageal resection. Correlated variables identified by chi-squared and Fisher‘s tests were included in the final analysis. A type I statistical error *p*-value of <0.05 was considered statistically significant.

## 3. Results

### 3.1. Patient Characteristics

Twenty-two (3.5%) patients who experienced an EAF were enrolled for analysis. Twenty-one (95.4%) of these underwent surgery specifically for EAF after esophagectomy and one in this group underwent surgery for EAF due to Hodgkin’s disease (#14). Another patient in this group underwent surgery due to EAF after esophagectomy and, two years later, a procedure was performed for a fistula to the lower right lobe (#10). 

In the presented analysis, one patient underwent surgery using the minimally invasive esophagectomy (MIE) technique (Ivor Lewis), and one had the McKeown procedure. The remaining patients underwent open esophageal resection using the Ivor Lewis method. 

### 3.2. Characteristic of the Study Group

Overall, 22 (3.5%) patients were included in the study. The mean age was 48.6 years (range: 18–78). There were no differences in demographic data (age, gender, ASA, BMI) in the study group (*p* = 0.22). The size of the tumor was 4.1 cm (range: 3.3–5.2 cm) and was not a significant risk factor for an EAF (*p* = 0.28). Twenty (90.1%) patients received preoperative radio-chemotherapy (according to the CROSS protocol) and two (9.1%) underwent chemotherapy (FLOT protocol). Radio-chemotherapy was not a significant risk factor for fistula formation after esophageal resection (HR: 0.02, 95%CI: −0.006–0.06, *p* = 0.12). During esophageal resection, patients had 18–41 lymph nodes removed, with an average of 21.3. The removed lymph nodes were not a risk factor for the occurrence of a fistula (*p* = 0.16). The duration of the procedure ranged from 273 to 456 min and did not have an impact on the occurrence of the fistula (*p* = 0.29).

In the study group, 21 (95.4%) patients developed an EAF, and one (4.5%) developed a bronchial fistula without an esophageal–gastric anastomotic leak. Distribution of the intrathoracic fistulae was as follows:thirteen (59.1%) patients had a bronchial fistula in the left main bronchus,seven (31.2%) developed a tracheal fistula,one (4.5%) patient developed a fistula to the right bronchus,one (4.5%) patient developed a fistula to the right lower lobe

The size of the fistulas ranged from 15 to 30 mm (mean 22 mm).

### 3.3. Surgical Treatment and Outcomes (Table 1)

#### 3.3.1. Primary Fistula Treatment without Esophageal Diversion

In one patient (4.5%), surgical treatment consisted of a primary plasty of the esophageal–gastric anastomosis and repair of the bronchial fistula with an IMF with good results (#15). One patient (4.5%) underwent re-anastomosis and upper right lobectomy with good results (#22). One (4.5%) underwent plasty using the LDM; however, the fistula persisted during the postoperative course. Esophageal stenting was then performed with a slight improvement in clinical condition. The patient ultimately required enteral feedings (#13). In one patient (4.5%), primary resection of the esophagus and tracheal plasty with a GOF was used with good results (#15). One patient (4.5%) underwent plasty with an IMF of the right main bronchus with good healing; however, during the postoperative period, there was a recurrence of the fistula at the esophageal anastomosis. The patient was subsequently treated with stenting (#11).

**Table 1 healthcare-11-03165-t001:** Clinical–pathological analysis of patients with airway fistulas.

No/Sex/Age	Indication for Esophagectomy	Type of Fistula	Location of Damage of the Bronchial Tree	Diagnosis of Damage Time	Type of Surgical Repair	Restoration of Alimentary Tract	Results	Post Operative Survival/Days
1/M/52	SCC	BF type II	LMB	PD (6)	Stenting of LMB	-	Death	
2/M/58	SCC	BF type II (suspected)	LMB	PD (10)	Conservative	-	Good	269
3/M/68	SCC	BF type II	LMB	PD (7)	Suture of LMB and ICF plastic DE	Colon	Good	485
4/M/72	SCC	BF type II	RMB	PD (7)	Suture of RMB and plastic IMF, DE	No restoration due to dissemination	Bad	268
5/M/65	SCC	BF type II	LMB	PD (7)	Suture of LMB and IMF, DE	Colon	Good	895
6/M/63	SCC	BF type II	RMB	PD (8)	Suture of RMB and IMF, DE	Colon	Good	528
7/M/69	SCC	BF type II	LMB	PD (7)	Plastic with biomaterial, DE	Colon	Good	579
8/M/74	AEG	BF type II	LMB	PD (10)	Plastic with biomaterial, DE	No restoration due to dissemination	Good	295
9/M/68	SCC	BF type II	LMB	PD (8)	Suture of RMB and IMF, DE	Colon	Good	285
10/M/55	SCC	TF type II, F-LRL type IV	T, LRL	PD (7)	Suture of T and PMM plastic. After 2 years, hemorrhage, resection of the gastric graft, DE. Wedge resection of the lung parenchyma with fistula of the LRL.	Colon	Good	369
11/M/68	AEG	BF type I	RMB	PD (8)	Primary suture of RMB and IMF, anastomosis plastic, recurrence of esophageal fistula, stenting	-	Bad	85
12/F/63	SCC	TF type II	T	PD (7)	Suture of LMB and IMF, DE	Colon	Good	469
13/M/50	SCC	TF type I	T	PD (8)	Primary treatment	LDM	Bad—fistula recurrence	356
14/F/17	HL	TF	T	Stenting injury	Primary McKeown operation, suture of trachea and GOP	Gastric conduit	Good	Alive
15/M/69	SCC	BF type I	LMB	PD (9)	Primary suture of esophago-gastric anastomosis, myoplasty of the fistula IMF	Gastric conduit	Good	725
16/M/64	SCC	TF type II	T	PD (12)	Suture of fistula myoplastic with IMF, DE	Colon	Good	586
17/M/62	SCC	BF type II	LMB	PD (11)	Suture of fistula myoplastic with IMF, DE	Colon	Good	489
18/F/52	AEG	BF type II	LMB	PD (7)	Suture of fistula myoplastic with ICF, decortication of the right lung, DE	Colon	Good	701
19/M/58	SCC	BF type II	LMB	PD (12)	Suture of fistula myoplastic with IMF, DE	Colon	Good	735
20 20 M/68	SCC	BF type II	LMB	PD (10)	Suture of fistula myoplastic with IMF, DE	Colon	Good	690
21/M/55	SCC	BF type II	LMB	PD (8)	Suture of fistula myoplastic with IMF, DE	Colon	Good	Alive
22/M55	SCC	BF type II	URB	PD (12)	Primary esophago-gastric re-anastomosis, URL	-	Good	Alive

AEG—adenocarcinoma of the esophago-gastric junction, BF—bronchial fistula, DE—diversion of the esophagus, F-LRL—fistula of the lower right lobe, GOP—gastric omental plastic, HL—Hodgkin’s Disease, IMF—intercostal muscle flap, LDM—latissimus dorsi muscle, LRL—lower right lobe, LMB—left main bronchus, PMM—pectoralis major musculus, PD—postoperative day, RMB—right main bronchus, SCC—squamous cell carcinoma, T—trachea, TF—tracheal fistula, URB—upper right bronchus, URL—upper right lobectomy.

#### 3.3.2. Secondary Fistula Treatment with Diversion of the Esophagus

Fifteen (68.1%) of the patients had a diversion of the esophagus. Of these, 13 (59.1%) underwent tracheobronchial plasty using an IMF, 2 (9%) using a biomaterial (Figure 3), 1 (4.5%) using the LDM, and 1 (4.5%) using the PMM. 

#### 3.3.3. Stenting

One patient (4.5%) was treated with stenting of the left main bronchus but unfortunately passed away due to sepsis and multiorgan dysfunction (# 1).

#### 3.3.4. Conservative Treatment

In one patient (4.5%), a bronchial fistula with the ulcer of an esophago–gastric anastomosis was treated conservatively with good results (# 2).

### 3.4. Bacterial Characteristics and Septic Threats

In the group of treated patients, seven (31.8%) were found to have features of progressive mediastinitis, and four (18.9%) had limited mediastinitis. Eight patients required lung decortication, and three (13.6%) were diagnosed with a mediastinal abscess (Table 2 and Table 3). Patients underwent bronchoscopy with bronchoalveolar lavage (BAL) and pleural fluid was collected in order to identify the bacterial flora (Table 4). Re-operated patients with type II and III fistulas had positive cultures in microbiological tests. In the post-operative course, patients received antibiotic therapy according to the obtained cultures. 

### 3.5. Postoperative Management

Complications after treatment are presented in Table 5. Fourteen (63.6%) patients required mechanical ventilation for 3 to 33 days (mean 9 days). ICU stays ranged from 5–63 days, with a mean of 17 days. 

### 3.6. Restoring the Continuity of the Gastrointestinal Tract

In 13 patients (62.5%), the continuity of the gastrointestinal tract was restored using the colon (retrosternal route), while in 2 patients (9.0%), the attempted procedure was abandoned due to tumor dissemination. The comparison of results and treatment methods by other authors is presented in Table 6.

#### Follow-Up

The mean follow-up time was 26.8 months (range 3–86). Twenty-one patients (83.3%) were evaluated postoperatively; three (12.5%) were lost during follow-up. No recurrence of the fistula was found during the follow-up, and patients were free of other intrathoracic complications with good clinical condition. Post-operative mean survival was 463.6 days (14–895). The survival of the patients is given in the Table 1.

## 4. Discussion 

EAF after esophageal resection is a relatively rare complication that poses an immediate life-threatening risk and is associated with a mortality rate reaching up to 57% [11,12,16]. In current literature, authors often present relatively small groups of treated patients, for whom the approach varies, underscoring the complexity of the issue at hand (Table 6).

An important element of describing a patient’s clinical condition, referring to it and comparing it to the results of other authors, is the classification of EAF after esophagectomy. In the cited literature, the classification of EAF was developed by Yasuda et al., with a focus on the leakage of the esophageal graft and the penetration of the ulcer or fistula into the bronchial tree [17]. Bartels et al. also contributed to this classification by highlighting the ischemic zone that may occur after esophageal resection within the bronchial tree, which predisposes patients to the occurrence of a fistula [18]. The classification proposed by Wang et al. has practical implications [12]. In our classification, we reference detailed topographic locations and clinical significance assessing the possibilities of surgical treatments. 

The pathogenesis of an EAF has not been fully explained. Maruyama et al. postulated that an extensive lymphadenectomy with excision of approximately 60 lymph nodes can lead to TBT devascularization, which can promote fistula development [16]. Similarly, life-saving surgery after radical radio-chemotherapy is a risk factor for fistula development at a rate of about 7% [19]. Mechanical injury to the TBT caused by the stapler suture line is a rare cause as confirmed in our report. One of the most important causes of fistula complications is neoadjuvant radio-chemotherapy. Wang et al. confirmed in a large group of patients undergoing esophagectomy (945 cases) that it was the main cause of TBT ischemia and subsequent fistula creation [20]. Our research does not confirm the observations of these authors. Wang et al. also indicated additional risk factors for fistula creation such as an anastomotic leak, posterior mediastinal route, and an anastomotic site in the vicinity of the TBT [20]. Among the potential causes of secondary fistulas, one should consider the technical aspect of the instruments used during esophageal resection. The instruments that can generate high temperatures especially pose a risk for postoperative fistula formation. 

In the current study, leakage at the esophageal anastomosis was accompanied by a fistula in the bronchial tree with a diameter between 15 and 30 mm. It was diagnosed between postoperative days 6 to 12 (mean 8.5). A fistula in the bronchial tree with a diameter of up to 2 cm was repaired successfully in 13 patients using an IMF for plasty. In defects greater than 2 cm, a flap from the LDM, GOF, PMM, or a biomaterial was used (Table 2). Closure with a bio-prosthetic patch (Permacol, Covidien) was used in two patients. Adequate tissue ingrowth into the prosthetic patch and fistula closure were achieved in both patients. Similarly, Udelsman et al. recommended this treatment option for cases involving large tissue defects [21].

Among the patients who underwent surgical treatment, healing of bronchial tree fistulas was achieved in 21 (95.4%) patients. Rosskopfova et al. reported a similar cure rate of 95% for bronchial tree fistulas using LDM plasty. The use of an LDM flap is a very effective method, especially in large defects of the TBT [10]. 

While the treatment for a bronchial tree fistula is effective, anastomotic leakage is a major problem. The degree of ischemia in the area of the leaking anastomosis is difficult to evaluate, and the treatment undertaken is often ineffective resulting in fistula recurrence and a complicated postoperative course. Hence, the esophageal diversion is one of two essential components of our management. An early esophageal diversion was performed in 17 patients (70.8%). Treatment of EAF in our report is associated with a high percentage of perioperative complications but an acceptable postoperative mortality. Among the analyzed patients, 14 (63%) experienced severe postoperative complications requiring intensive care, and one (4.5%) passed away after bronchial stenting due to multi-organ failure (Table 1 and Table 5).

In a study by Rosskopfova et al., esophageal diversions were performed in eight patients (36.3%). In three patients who underwent esophageal fistula repair, the fistula recurred, and one patient died due to a hemorrhage resulting from an intestinal leak [10]. The authors concluded that performing a myoplasty for a leaking esophageal anastomosis is a risky procedure. Wang et al. also reported a high mortality rate (42%) in patients with a type II fistula who did not undergo an esophageal diversion, highlighting the technical difficulties associated with the treatment of EAF using a gastric graft left behind [12]. Palmes et al. also reported the ineffectiveness of the procedure in eight patients who underwent stenting or early re-anastomosis, with a mortality rate of 53% [15]. A promising approach is presented by Bertheuil et al., who used a perforator-based intercostal artery muscle flap in eight treated patients with EAF following esophageal resection. The author reported a recurrence of a fistula in two patients with no perioperative mortality [13]. 

Restoration of gastrointestinal continuity was performed after three months in 13 patients using displaced colon to complete an esophageal anastomosis at the neck. In the preoperative period, these patients were fed an enteral diet. A satisfactory postoperative quality of life was achieved in these patients. The restoration of gastrointestinal continuity is not always possible, which is influenced by the progression of the cancer and the patient’s clinical condition. This emphasized by Balakrishnan et al., who reported that the restoration of gastrointestinal continuity was possible in two out of five patients [9]. 

Among other management options, stenting of the TBT and possible anastomotic leaks are also suggested. Qualification for this treatment approach is challenging because there are no clear guidelines for its implementation. Wang et al. treated 58 patients and were able to achieve healing in 20 patients and a complete seal in 45 patients, reducing the complication risk of sepsis [22]. Lambertz et al. and Palmes et al. suggest that in patients of limited mediastinitis, interventional endoscopic techniques should be employed, reserving esophageal diversion only for persistent mediastinitis [3,15]. Among our treated patients, two underwent unsuccessful stenting procedures. Unfortunately, one patient passed away, and in the case of the other patient, stenting alleviated the symptoms of the fistula, but enteral nutrition was maintained. Maruyama et al. suggested that this type of treatment is ineffective, as it can be difficult to span the whole defect with a stent, and stenting the anastomotic leak can be precarious [16]. However, we believe stenting should be reserved for a narrow group of patients.

Patients with AEF are particularly susceptible to the development of mediastinitis and multi-organ-failure [3,12,14,15]. Among treated patients, eight (36.4%) were diagnosed with diffuse mediastinitis (seven with type III fistula and one with type II), while its limited form was observed in four (18.9%) patients with type II fistula. Our approach included the primary management of AEF in patients with type II fistula and the esophagogastric re-anastomosis, which is challenging, not always feasible, and associated with uncertain postoperative outcomes [13]. In type III fistulas, the principle adopted is the management of the fistula in the bronchial tree by performing myoplasty or implanting biomaterial with esophageal diversion. Patients with a type III fistula are particularly at risk, with an approximate 60% incidence of severe complications. In our approach, the mortality rate was 4.5%, and continuity of the digestive tract could be restored in 13 patients. This approach minimizes the occurrence of mediastinitis symptoms and septic complications, particularly in patients with Type 3 fistulas. The risk of lethal complications is confirmed by results of Lambertz et al., Wang et al., and Palmes et al., where the mortality rates were 38%, 42%, and 47% respectively [3,10,11]. 

In addition, conservative management may be appropriate for patients in whom the fistula is covered with an esophageal graft, and a CT scan reveals fully expanded lungs. These patients require close monitoring, chest drainage, antibiotic treatment, jejunal nutrition, and bronchoscopy. 

The presented paper has some limitations. The limitations include that it was a retrospective analysis, had a small sample size, and included a diverse group of patients receiving the proposed treatments. Despite these limitations, our study shows that esophageal diversion allows for a safe and successful plasty of bronchial fistulas using an IMF without unnecessarily using muscle displacement with an extrathoracic technique or free flap. Our results support recommending the use of an intercostal flap for bronchial fistula repair.

## 5. Conclusions

In summary, this approach provides full control of the leak at the esophageal anastomosis and the septic process within the mediastinum, which determines patient survival. In all operated patients, we used pedunculated flaps without the need for free flaps. We believe that this treatment technique in conjunction with the early diagnosis of EAFs should always be considered.

## Figures and Tables

**Figure 1 healthcare-11-03165-f001:**
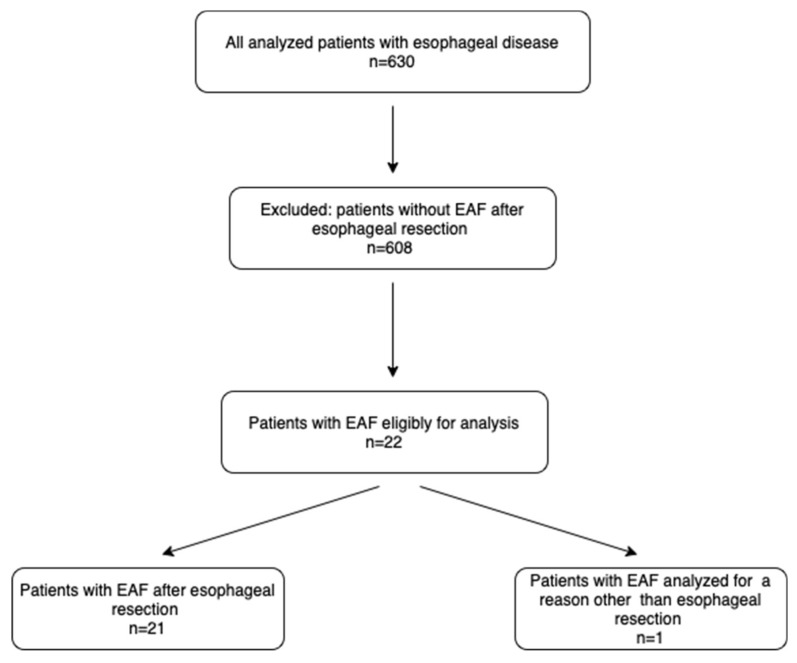
Flowchart for recruitment of patients for the study. EAF: esophago–airway fistula.

**Figure 2 healthcare-11-03165-f002:**
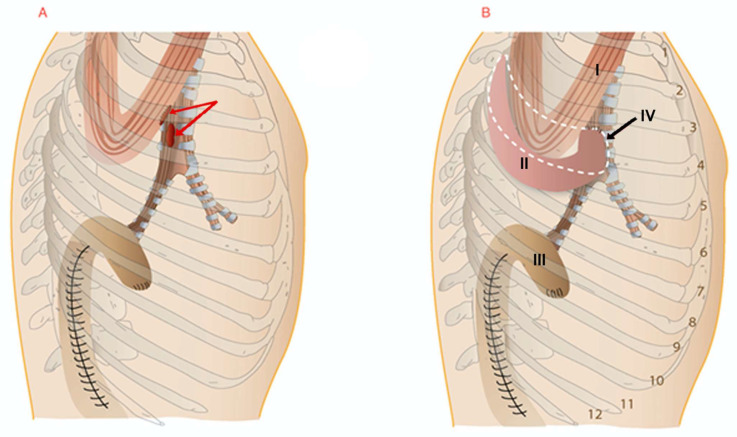
(**A**) Scheme showing an esophago-airway fistula after esophagectomy. The red arrows indicate the esophago-airway fistulae. (**B**) Airway plastic fistula (type III) using an intercostal muscle flap (IMF) with diversion of the esophagus. I—the esophagus is prepared for diversion; II—the IMF sutured into the airway fistula; III—separated gastric conduit; IV—dotted area shows the plane of IMF displacement. Ribs are numbered.

**Figure 3 healthcare-11-03165-f003:**
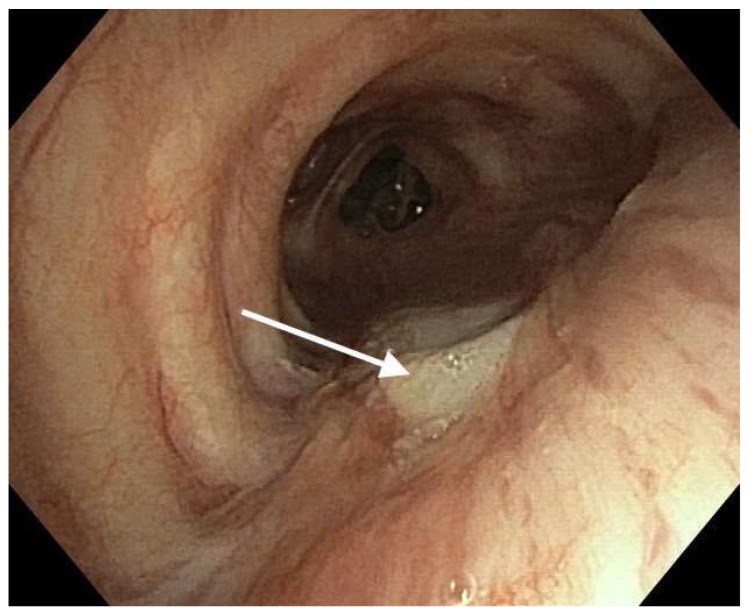
Reconstruction of the airway fistula sewing a biomaterial (Permacol) into the left main bronchus. The arrow indicates embedded biomaterial in the bronchial wall.

**Table 2 healthcare-11-03165-t002:** Septic characteristics in patients with EAF.

Fistula Type	Decortications	Abscess	Empyema	Mediastinitis	
				L	P
Type II	1	0	1	4	1
Type III	7	3	7		7

EAF—esophago–airway fistula, L—limited mediastinitis, P—progressive mediastinitis.

**Table 3 healthcare-11-03165-t003:** Postoperative complications in patients with EAF.

Fistula Type	Respiratory Insufficiency	Sepsis Shock	Circulatory Insufficiency	Renal Insufficiency	Multiorgan Dysfunction	Death
Type II	1	1	3	0		
Type III	3	3	5	2	1	1

**Table 4 healthcare-11-03165-t004:** Bacterial colonization in operated patients.

Type of Tested Material	Type of Microorganism	Number of Isolates
Bronchoalveolar lavage (BAL)	*Physiological flora*	7 (31.8%)
	*Candida albicans*	4 (18.2%)
	*Streptococcus pneumoniae*	8 (36.4%)
	*Escherichia coli*	5 (22.7%)
	*Pseudomonas aeruginosa*	5 (22.7)
	*Proteus vulgaris*	2 (9.1%)
Pleural cavity fluid	*Physiological flora*	2 (9.1%)
	*Candida albicans*	7 (31.8%)
	*Candida glabrata*	5 (22.7%)
	*Streptococcus pneumoniae*	7 (31.8%)
	*Escherichia coli*	7 (31.8%)
	*Pseudomonas aeruginosa*	8 (36.4%)
	*Proteus vulgaris*	5 (22.7%)
	*Klebsiella pneumoniae*	6 (27.3%)
	*Klebsiella pneumoniae (ESBL)*	2 (9.1%)
	*Staphylococcus aureus*	5 (22.7%)
	*Moraxella catharalis*	2 (9.1%)
	*Enterococcus faecium*	2 (9.1%)
	*Corynebacterium* sp.	4 (18.2%)
	*Enterococcus faecalis (VRE)*	2 (9.1%)

ESBL: extended-spectrum beta-lactamases; VRE: Vancomycin-Resistant Enterococcus.

**Table 5 healthcare-11-03165-t005:** Surgical complications according to the Clavien–Dindo classification [8].

Grade	Number (%)	Definition
I	17 (72.3%)	Urinary tract infection—7 (31.8%); wound infection—3 (13.6%); temporary mental disorders (postoperative delirium)—5 (22.7%) Any deviation from the normal postoperative course without the need for pharmacological treatment or surgical, endoscopic, and radiological interventions. Allowed therapeutic regimens are drugs such as antiemetics, antipyretics, analgetics, diuretics, and electrolytes and physiotherapy. This grade also includes wound infections opened at the bedside.
II	6 (27.2%)	Pneumonia—4 (18.1%), Nervus laryngeus paresis—2 (9.0%) Requiring pharmacological treatment with drugs other than those allowed for grade I complications. Blood transfusions and total parenteral nutrition are also included.
III	14 (63.6%)	Patients requiring bronchoscopy—8 (36.4%), tracheostomy—6 (27.2%) Requiring surgical, endoscopic, or radiological intervention.
IIIa		Intervention not under general anesthesia.
IIIb		Intervention under general anesthesia.
IV	8 (36.3%)	Respiratory insufficiency—4 (18.1%), Septic shock—4 (18.1%) Respiratory insufficiency. Life-threatening complication (including CNS complications); * requiring IC/ICU-management
IVa	6 (27.2%)	Single organ dysfunction (including dialysis) Renal insufficiency—2 (9.0%), Circulatory insufficiency—4 (18.1%)
IVb	1 (4.5%)	Sepsis—1 (4.5%) Multiorgan dysfunction.
V	1 (4.5%)	1 (4.5%) Death of a patient.

* Brain hemorrhage, ischemic stroke, subarachnoidal bleeding, but excluding transient ischemic attacks. CNS—central nervous system, IC—intermediate care, ICU—intensive care unit.

**Table 6 healthcare-11-03165-t006:** Results and management of patients treated with EAF after esophageal resection.

Reference Author	Year	No. of Patients after Esophagectomy	EAF after Other Esophageal, Mediastinal, or Other Disease	EAF after Airway Tract Resection	Primary Closure	Diversion of the Esophagus	Type of Flap Used for Reconstruction or Stenting	EAF Postoperative Mortality	Restoring the Continuity of Digestive Tract	EAF Recurrence	No. of Patients Who Survived (%)
Balakrishnan et al. [9]	2018	11	none	none	3 (27.3%) and 1 pr	7 (63.6%)	PMF	3 (27%)	2 (18.2%)	2 (18.2%)	3 (27.3%)
Rosskopfova et al. [10]	2017	22	5	2	5 (22.7%)	9 (40.9%)	LDF, MSAF, PMF	4 (18%)	5 (22.7%)	2 (4.5%)	18 (81.8%)
Shweigert et al. [11]	2012	7	none	none	1 (14.3%) (ct)	2 (28.6%)	Stent, MSA,	2/2 (28.6%)	none	2 (28.6%)	2 (28.6%)
Lambertz et al. [3]	2016	13	none	none	3 (27.3%) after stenting,	2 (63.6%)	Stenting, PPF, SCMM,	5 (38.5%)	unknown	2 (18.2%)	7 (63.6%)
Wang et al. [12]	2020	26	none	none	4 (15%) (ct)	none	PSFF, IMF, stent,	11 (42.3%)	none	1 (3.8%)	15 (57.7%)
Bertheuil et al. [13]	2021	8	none	none	none	0	pIMF	0	8 (10)	2 (25%)	8 (100%)
Fricke et al. [14]	2017	13	2	4	0	1 (7.7%)	IMF, PMF, LDF, RAMF, FTPFF	0	13(100%)	5 (38.5%), after re-op 1	13 (100%)
Palmes et al. [15]	2021	15	none	none	3 (ct)	4 (26.7%)	Stent, ps	7 (47%)	unknown	1 (6.7%)	7 (46.7%)

ct, conservative treatment; FTPFF, free temporo–parietal fascia flap; IMF, intercostal muscle pedicled flap; LDF, latissimus dorsi pedicled flap; MSAF, musculus serratus anterior flap; IMF, perforator-based intercostal artery muscle flap; PMF, pectoralis major pedicled flap; PPF, pediculated pericardial flap; PSFF, pedicled subcutaneous fascia flap; pr, primary restoring; ps, primary suture; RAMF, rectus abdominis myocutaneous flap; SCMM, sternocleidomastoideus mascule.

## Data Availability

Data are available upon request.

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
