# Peer review of "Treatment of Esophago–Airway Fistula after Esophageal Resection"

_healthcare, 2023, doi:10.3390/healthcare11243165_

Round 1

Reviewer 1 Report

Comments and Suggestions for Authors

Fistulas between the upper intestinal tract and the airway post-esophagectomy are rare. They are a severe complication with significant mortality. Treatment and therapy are challenging and require careful multidisciplinary management. This is indeed an important study that provides alternative novel approaches to managing the EOF post-esophagectomy. My comments are as follows:

1.      In the abstract, it is noted that 21 patients experienced a tracheobronchial fistula and one was treated for different reason. Please mention this clinical reasoning in the abstract and in your text where relevant.

2.      In the materials and methods section, please provide a detailed outline of inclusion and exclusion criteria for this retrospective study in the text.

3.      In the methods section, “Patients diagnosed with Type I fistula without signs of ischemia in the esophageal graft they were considered candidates for re-anastomosis and fistula repair in the bronchial tree using intercostal flap, pectoral, latissimus dorsi muscles or the insertion of biomaterial”. Please mention the biomaterials used.

4.      It is mentioned that one of the variables that was analyzed was the influence of pre-operative radio-chemotherapy on the occurrence of fistula. Was the dosage of radio-chemotherapy standard in all patients? Was the duration of radio-chemotherapy similar in all patients?

5.      Did any patient have a past medical history of other fibrotic pathologies of the esophagus?

6.      Was time to fistula as well recorded? If so please present this.

7.      Did the pre-operative size of the tumor have any effect on outcomes based on different methods/procedures used in the post-operative phase?

8.      Was duration of surgery ascertained and correlated with outcomes ?

9.      How did the hospitalization stay vary for different surgical methods used.

Author Response

Reviewer 1

Fistulas between the upper intestinal tract and the airway post-esophagectomy are rare. They are a severe complication with significant mortality. Treatment and therapy are challenging and require careful multidisciplinary management. This is indeed an important study that provides alternative novel approaches to managing the EOF post-esophagectomy. My comments are as follows:

  1. In the abstract, it is noted that 21 patients experienced a tracheobronchial fistula and one was treated for different reason. Please mention this clinical reasoning in the abstract and in your text where relevant.

 Corrected the text accordance with the reviewer’s suggestions

  1. In the materials and methods section, please provide a detailed outline of inclusion and exclusion criteria for this retrospective study in the text.

Corrected the text accordance with the reviewer’s suggestions

  • Inclusion and exclusion criteria were as follows:

The patient’s inclusion criteria:

  1. Patients qualified for radical treatment of esophageal cancer according to the CROSS or FLOT protocol and confirmed endoscopically (EUS, EBUS) with no invasion into the bronchial tree.
  2. Patients after radical esophagectomy R0.
  3. Patients with an endoscopically confirmed esophago-airway (EAF) fistula.
  4. Patients undergoing chemoradiotherapy for a different cancer than esophageal cancer and esophageal surgery resulting in an esophago-airway fistula.

The patient’s exclusion criteria:

  1. Patients after non-radical esophageal resection with an R1 feature.
  2. Patients after intraoperative injury to the bronchial tree.
  3. Patients with confirmed leakage only in the esophagogastric anastomosis.
  4. Patients with a preoperatively confirmed EAF in the course of esophageal cancer.
  5. Severe patient condition disqualifying them from surgical treatment.
  6. In the methods section, “Patients diagnosed with Type I fistula without signs of ischemia in the esophageal graft they were considered candidates for re-anastomosis and fistula repair in the bronchial tree using intercostal flap, pectoral, latissimus dorsi muscles or the insertion of biomaterial”. Please mention the biomaterials used.

Corrected the text accordance with the reviewer’s suggestions

  1. It is mentioned that one of the variables that was analyzed was the influence of pre-operative radio-chemotherapy on the occurrence of fistula. Was the dosage of radio-chemotherapy standard in all patients? Was the duration of radio-chemotherapy similar in all patients?

Yes, the procedure was standard according to the CROSS and FLOT protocol. Corrected the text accordance with the reviewer’s suggestions

  1. Did any patient have a past medical history of other fibrotic pathologies of the esophagus?

Yes, one patient with Hodgkin’s disease, who developed high stricture of the esophagus after radiotherapy subsequently developed EAF following stent placement (app. 20cm from the incisors.

  1. Was time to fistula as well recorded? If so please present this.

Yes, correction in the text 

  1. Did the pre-operative size of the tumor have any effect on outcomes based on different methods/procedures used in the post-operative phase?

Thank you for suggestion, the correction in the text

The size of the tumor was 4,1cm range (3,3 – 5,2cm) and was not a significant risk factor for an EAF (p=0.285). 

  1. Was duration of surgery ascertained and correlated with outcomes?

Pierwotnie nie, korekta w tekÅ›cie. Czas leczenia nie miaÅ‚ wpÅ‚ywu na wystÄ…pienie powikÅ‚aÅ„. 

Thank you for suggestion, the correction in the text

The duration of the procedure ranged from 273 to 456 minutes and did not have an impact on the occurrence of the fistula p=0,29.

  1. How did the hospitalization stay vary for different surgical methods used.

In the presented analysis, 1 patient underwent surgery using the MIE technique (Ivor-Lewis). The remaining patients had open esophageal resection performed by the Ivor-Lewis method, and 1 had the McKeown procedure. Hospital stay did not differ due to the adopted surgical technique."

Corrected in the text.

Reviewer 2 Report

Comments and Suggestions for Authors

Authors deserve congratulations for their extensive work on patients undergoing esophageal resection and the management of esophago-tracheal and esophagobronchial fistulas. However, several concerns have been identified in the study:

1- The authors presented a significant series of esophago-bronchial fistulas, but it remains unclear if all patients had tracheo-bronchial fistulas. Additionally, the specific locations of esophageobronchial fistulas (e.g., main bronchus, intermediary bronchus) are not specified.

2- If the authors performed parenchymal resection, such as lobectomy due to fistula, it should be clearly documented.

3- On page 4, the criteria for initiating conservative treatment raise questions. The meaning of "distended parenchyma" and the clarification of "gastric or intestinal graft covering the fistula" need to be provided. Did the authors use intestine or stomach to cover the bronchus?

4- In the definition of the IMF technique, the statement about puncturing the bronchial wall needs further explanation. Did the authors puncture the bronchus first, and why? This step should be clarified.

5- The use of biomaterials and suturing techniques should be explained in more detail for better understanding.

6- The mean follow-up time and the pathologies of the resections should be described to provide a comprehensive overview.

7- Individual patient definitions in the Results section should be avoided. Instead, all results should be summarized categorically.

8- Complications, occurrences, and their percentages should be presented in a table for better clarity.

9- The definitions for 'Good' and 'Bad' results in Table 3 should be explicitly stated. What does a 'Bad' result mean?

10- In Table 3, clarification is needed on whether the 'survival' parameter represents 'follow-up time' rather than time to death.

11- Considering mediastinitis as a crucial complication, the authors are expected to define their protocol for managing mediastinitis in their series.

12- The Discussion section should focus solely on the interpretation and discussion of results. Results, such as those presented in Table 4, should be appropriately placed in the Results section.

13- The fate of feeding jejunostomies should be described, including when authors closed those jejunostomies.

14- In minor comments, it is suggested to include the brands of sutures along with the material used.

15- Percentages should be included in Table 1 for a more informative presentation of the data.

In conclusion, it is recommended that the authors address these concerns and provide additional clarity to enhance the overall comprehensibility of the study.

Comments on the Quality of English Language

It should be improved.

Author Response

Reviewer 2

Authors deserve congratulations for their extensive work on patients undergoing esophageal resection and the management of esophago-tracheal and esophagobronchial fistulas. However, several concerns have been identified in the study:

  • The authors presented a significant series of esophago-bronchial fistulas, but it remains unclear if all patients had tracheo-bronchial fistulas. Additionally, the specific locations of esophageobronchial fistulas (e.g., main bronchus, intermediary bronchus) are not specified.

Yes, the text provided location of the fistulas. The correction in the text

Distribution of the intrathoracic fistulae:

  1. thirteen (59.1%) patients had a bronchial fistula in the left main bronchus,
  2. seven (31.2%) developed tracheal fistula,
  3. one (4.5%) patient developed fistula to the right bronchus,

one (4.5%) patient developed fistula to the right lower lobe   

  • If the authors performed parenchymal resection, such as lobectomy due to fistula, it should be clearly documented.

The correction in Table 1.

Suture of T and PMM plastic. After 2 years, hemorrhage, resection of the gastric graft, DE. Wedge resection of the lung parenchyma with fistula of the LRL.

  • On page 4, the criteria for initiating conservative treatment raise questions. The meaning of "distended parenchyma" and the clarification of "gastric or intestinal graft covering the fistula" need to be provided. Did the authors use intestine or stomach to cover the bronchus?

The evaluation in the performed tomography and in the endoscopic examination (bronchoscopy) allows assessing the size of the fistula and its coverage with an esophageal graft. An essential aspect of the assessment is the evaluation of air leakage in the pleural drainage; if present, the patient is qualified for surgical treatment and bronchial fistula repair. In the absence of air leakage and with expanded lung parenchyma, the patient is eligible for conservative treatment. The coverage of the bronchial fistula depends on the initial choice of graft, most commonly the stomach, and less frequently the large intestine.

Conservative treatment was initiated in patients who had a fistula of up to 1 cm in size, no pleural fluid on CT scan, distended pulmonary parenchyma without pneumothorax, air leakage and a gastric graft covering the fistula. All patients were started on broad-spectrum antibiotic therapy. Control bronchoscopies were performed on postoperative days 1, 3, 5, and 7.

Corrected

  • In the definition of the IMF technique, the statement about puncturing the bronchial wall needs further explanation. Did the authors puncture the bronchus first, and why? This step should be clarified.

The bronchial wall was then punctured close to the fistula from the outside to the inside, then approximately 2 mm from the puncture, a suture was pierced outward, and the muscle was punctured from the inside to the outward. After this suture was placed, the muscle was slipped over the injured bronchus and the sutures were tied. Three to four stitches were placed.

Additional explanations of the surgical technique has been introduced in the text

  • The use of biomaterials and suturing techniques should be explained in more detail for better understanding.

The biomaterial Permacol (Covidien) were used and was sewn into the fistula using single or running sutures and then reinforced with an IMF by use interrupted sutures PDS 000.

The recommended correction has been introduced in the text

  • The mean follow-up time and the pathologies of the resections should be described to provide a comprehensive overview.

The mean follow-up time was 26.8 months (range 3-86). Twenty-one (83.3%) of them were evaluated postoperatively, 3(12.5%) patients lost during follow-up. No recurrence of the fistula was found during the follow-up patients were free of other intrathoracic complications with good clinical condition. Post operative mean survival was 463.6 (14-895). The survival of the patients is given in the Table 1.

The recommended correction has been introduced in the text

  • Individual patient definitions in the Results section should be avoided. Instead, all results should be summarized categorically.

The recommended correction has been introduced in the text

  • Complications, occurrences, and their percentages should be presented in a table for better clarity.

The recommended correction has been introduced in the text

  • The definitions for 'Good' and 'Bad' results in Table 3 should be explicitly stated. What does a 'Bad' result mean?

The report did not assess the quality of life of patients in detail. A good outcome was defined as the ability of patients to be orally nourished without the need for enteral nutrition. Conversely, a poor outcome was defined as the inability to orally nourish and the need for enteral nutrition. Patients for whom the restoration of gastrointestinal continuity was not possible were considered to have a poor outcome.

Thank you for suggestion. The recommended correction has been introduced in the text

(“Material and Methods”)

The restoration of gastrointestinal continuity was performer after 3 months, achieving the stabilization of the patient’s general condition. Until the time of surgical treatment, patients remained on enteral nutrition. For the purpose of this report, the patient’s quality of life after completing the surgical treatment was broadly defined by assessing the patient’s ability to nourish themselves. A good outcome was defined as the ability to oral feeding with meeting the energy requirements, while a poor outcome was defined as the inability for full oral nutrition or the inability to restore gastrointestinal continuity.

  • In Table 3, clarification is needed on whether the 'survival' parameter represents 'follow-up time' rather than time to death.

The correction was implemented in the table accordance with the reviewer’s recommendations

In the table, survival and time to death were included.

  • Considering mediastinitis as a crucial complication, the authors are expected to define their protocol for managing mediastinitis in their series.

Our approach is dependent on the type of fistula. Type 3 fistula posed a particular threat due to the anastomotic leak, inadequate blood supply to the esophageal graft, and fistula in the bronchial tree. In this type, we decided to exclude the esophagus. Although the procedure is challenging, and the patient's condition after treatment is difficult for them to accept, perioperative mortality is low.

The correction was implemented accordance with the reviewer’s recommendations

Mediastinitis has been defined as the presence of infected content in the mediastinum in the form of fluid and/or abscess, and/or necrotic tissues of the posterior mediastinum. A classification has been adopted distinguishing limited mediastinitis associated with type II fistula and progressive (diffuse) mediastinitis associated with type III. (Material and Methods)

Among our patients, 7 were diagnosed with progressive mediastinitis, and 4 with a limited form of it. Our approach included the primary management of AEF in patients with type II fistula and the esophagogastric re-anastomosis, which is challenging, not always feasible, and associated with uncertain postoperative outcomes (17). In type III fistulas, the principle adopted is the management of the fistula in the bronchial tree by performing myoplasty or implanting biomaterial with esophageal diversion. Patients with a type III fistula are particularly at risk, with an approximate 60% incidence of severe complications. In our approach the mortality rate was 4.5%, and continuity of the digestive tract could be restored by 13 patients.This approach minimizes the occurrence of mediastinitis symptoms and septic complications, particularly in patients with Type 3 fistulas. The risk of lethal complications is confirmed by results of Lambretz et al., Wang at al., and Palmes at al, where the mortality rates were 38%, 42%, and 47% respectively [3,7,15].

Table 2. Septic characteristics in patients with EAF

Fistula Type

Decortications

Abscess

Empyema

Mediastinitis

L              P

Type II

0

0

1

4

Type III

8

3

7

                  7

L – limited mediastinitis, P – progressive mediastinitis

Table 3. Postoperative complications in patients with EAF

Fistula Type

Respiratory insufficiency

Sepsis shock

Circulatory insufficiency

Renal insufficiency

Multiorgan dysfunction

Death

Type II

1

1

3

0

  • The Discussion section should focus solely on the interpretation and discussion of results. Results, such as those presented in Table 4, should be appropriately placed in the Results section.

The correction was implemented accordance with the reviewer’s recommendations

  • The fate of feeding jejunostomies should be described, including when authors closed those jejunostomies.

The jejunostomy is usually removed within 30 days after surgical treatment if the patient is nourishing orally. In case of difficulties in nutrition and recurrence of the fistula, we maintain enteral nutrition.

The correction was implemented accordance with the reviewer’s recommendations

  • In minor comments, it is suggested to include the brands of sutures along with the material used.

The correction was implemented accordance with the reviewer’s recommendations

(PDS000)

  • Percentages should be included in Table 1 for a more informative presentation of the data.

The correction was implemented in Table accordance with the reviewer’s recommendations.

In conclusion, it is recommended that the authors address these concerns and provide additional clarity to enhance the overall comprehensibility of the study.

Round 2

Reviewer 1 Report

Comments and Suggestions for Authors

All changes have been made by the authors in line with my recommendations making this a scientifically sound and detailed observation.

Author Response

Dear Editor-in-Chief van Zundert Andre, Prof

According to the reviewer’s recommendations corrections have been made in the paper. The changes have been highlighted in green. Reference to the changes has been made in both submitted documents (Word and PDF). I’m sending both documents.

I would like to inquire whether, in the event of the paper being accepted for publication, it can be submitted directly to editorial proofreading, or if I should handle this personally?

Please, if possible, submit the paper for editorial proofreading by MDPI.

Sincerely,
on behalf of all co-authors

Janusz WÅ‚odarczyk

jr.wlodarczyk@gmail.com

PS Unfortunatelly, I'm unable to attach the second document (PDF); I will send it via mail.
